# Fuel-Lubricant Interactions: Critical Review of Recent Work

Robert Ian Taylor

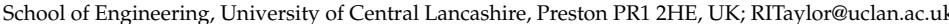

School of Engineering, University of Central Lancashire, Preston PR1 2HE, UK; RITaylor@uclan.ac.uk

**Abstract:** A critical review of recent work on fuel lubricant interactions is undertaken. The work focusses on liquid fuels used in diesel and gasoline vehicles. The amount of fuel that contaminates the lubricant depends on driving conditions, engine design, fuel type, and lubricant type. When fuel contaminates a lubricant, the viscosity of the lubricant will change (it will usually decrease), the sump oil level may increase, there may be a tendency for more sludge formation, there may be an impact on friction and wear, and low speed pre-ignition could occur. The increased use of biofuels (particularly biodiesel) may require a reduction in oil drain intervals, and fuel borne additives could contaminate the lubricant. The move towards the active regeneration of particulate filters by delayed fuel post-injection and the move towards hybrid electric vehicles and vehicles equipped with stop-start systems will lead to increased fuel dilution. This will be of more concern in diesel engines, since significant fuel dilution could persist at sump oil temperatures in the range of 100–150 °C (whereas in gasoline engines the more volatile gasoline fuel will have substantially evaporated at these temperatures). It is anticipated that more research into fuel lubricant interactions, particularly for diesel engines, will be needed in the near future.

**Keywords:** fuel; lubricant; fuel-lubricant interaction; fuel dilution; biofuels; friction modifiers

## 1. Introduction

Fuel/lubricant/engine interactions are not as important as fuel/engine or lubricant/engine interactions, and so are not as high on the list of priorities for either fuel scientists or lubricant scientists, unless field issues emerge.

It is well known that unburnt fuel, and fuel additives, can accumulate in lubricants, and in sufficient concentration can cause issues, such as (1) altering (usually lowering) the viscosity of the lubricant, (2) helping sludge to form in engines, (3) altering the oxidation properties of the lubricant (leading to lower oil drain intervals), and (4) potentially affecting the frictional properties of the lubricant.

In addition, the recent wider spread use of biofuels (mainly ethanol or methanol in gasoline fuel, or biodiesel in diesel fuel) has occasionally led to field issues (particularly in diesel engines).

Recent interest in fuel/lubricant interactions has surged due to (1) many more engines having stop-start systems, and (2) many engines having active aftertreatment systems in which extra fuel is injected (and burnt) to help with tailpipe emissions.

Fuel dilution is generally considered excessive if it exceeds 3–5%, although such levels can easily be reached with certain driving styles (many short trips from a cold start) and with modern vehicles that have aftertreatment systems with active regeneration.

This paper attempts to critically review work over the last 30 years, to give a fuller picture of what is currently known and what gaps remain.

## 2. Viscosity of Fuels

A recent paper by Yu et al. [1] reports kinematic viscosity measurements on passenger car diesel fuel, as detailed in Table 1 below.

**Table 1.** Recent kinematic viscosity measurements on passenger car diesel fuel from Yu et al. [1].

| Temperature (°C) | Kinematic Viscosity (cSt) |
|---|---|
| 40 | 3.08 |
| 60 | 2.16 |
| 80 | 1.62 |
| 100 | 1.28 |

Similar viscosity data on diesel fuel were also reported in an earlier paper by Tat and van Gerpen [2] and the variation of kinematic viscosity with temperature was fitted with a Vogel equation [3] (over the range 20–100 °C, the kinematic viscosity varied from approximately 4 cSt to 1.5 cSt, in broad agreement with the data in Table 1). This work was performed in the US, where there are very few diesel passenger vehicles, so the diesel fuel tested was most likely that for heavy duty vehicles.

Additional measurements of the viscosity of diesel fuels were reported by Schaschke et al. in 2013 [4]. These data also included measurements at high pressures. For measurements at atmospheric pressure, their measured dynamic viscosities ranged from about 3.2 mPa.s at 25 °C to 0.97 mPa.s at 100 °C. If fuel density is assumed to be approximately 0.8 g/cm$^3$, then these values would correspond to kinematic viscosities of 4 cSt at 25 °C and 1.2 cSt at 100 °C, which are in reasonable agreement with the work of Yu et al. [1] and Tat and van Gerpen [2].

Measurements on gasoline fuels have been reported by Trost et al. [5] and Table 2 summarizes their measured kinematic viscosity ($V_k$, cSt) and density data ($\rho$, g/cm$^3$) versus temperature (T, °C) for RON 95 and RON 98 gasoline fuels (RON is the research octane number; most standard fuels in Europe will be RON 95).

**Table 2.** Recent kinematic viscosity measurements on gasoline fuels from Trost et al. [4].

| T (°C) | $V_k$ (cSt) RON 95 Gasoline | $\rho$ (g/cm$^3$) RON 95 Gasoline | $V_k$ (cSt) RON 98 Gasoline | $\rho$ (g/cm$^3$) RON 98 Gasoline |
|---|---|---|---|---|
| −10 | 0.94 | 0.765 | 0.89 | 0.776 |
| 0 | 0.84 | 0.762 | 0.80 | 0.775 |
| 10 | 0.79 | 0.761 | 0.79 | 0.770 |
| 20 | 0.78 | 0.759 | 0.72 | 0.763 |
| 30 | 0.76 | 0.754 | 0.70 | 0.755 |
| 40 | 0.72 | 0.751 | 0.69 | 0.749 |

Other measurements of fuel kinematic viscosity and density were reported by Khuong et al. [6]. In this work, it was reported that the density of the RON 95 gasoline measured was 0.75 g/cm$^3$ at 15 °C and the kinematic viscosity was 0.542 cSt at 15 °C and 0.529 cSt at 20 °C.

In an interesting paper, Zhu et al. [7] reported on changes in viscosity and Reid vapour pressure that occurred due to evaporation of gasoline. The authors found that the dynamic viscosity of a RON 93 gasoline increased from around 0.9 mPa.s to 1.51 mPa.s when a sample of the gasoline was left exposed to the atmosphere for 30 days (it is implied that the sample was simply kept at room temperature, but the exact temperature was not reported, although a reasonable assumption would be that it was around 20–25 °C). The authors stated that the increase in dynamic viscosity occurred due to the evaporation of lower molecular weight components, and this was consistent with a weight loss of the sample (the initial weight of the sample was approximately 42.5 g, and after 30 days this had decreased to about 24 g).

The work of Zhu et al. [7] does raise an interesting question about viscosity measurements on fuels. How do researchers prevent the evaporation of the lighter ends of the fuel at elevated temperatures when measuring fuel viscosity, and are checks carried out to ensure the composition of the fuel at these elevated temperatures is the same as

that at lower temperatures? It would be expected that fuel viscosity would decrease with temperature, but the decrease would not be as great if some of the lower molecular weight components had evaporated at the higher temperatures. The effect of evaporation is expected to be more significant for gasoline, compared to diesel fuel. The effect of evaporation of fuel components at elevated temperatures was discussed by Costa and Spikes [8] in their investigations of the impact of ethanol (from fuel) on tribo-films.

One final paper that deserves inclusion in this section is the work of Riazi et al. [9] on the viscosity of liquid hydrocarbon mixtures. The authors provided simple relations that related the viscosity of liquid hydrocarbons to their refractive index and applied this method to a large number of liquid hydrocarbons (including octane, which is often used as a model fuel) to predict their viscosity and how it would decrease with temperature.

### 3. Fuel Dilution Levels in Vehicles

Many researchers have investigated the amount of fuel that enters the lubricant, and the amount depends not just on the fuel type and engine design, but also on how the vehicle is operated.

Kollman et al. [10] carried out a series of "SNAIL" field trials, in which a number of different gasoline cars were only ever driven short distances (up to 10 km) from a cold start. For this type of operation, fuel dilution in the sump oil was typically in the range 10–20% after 4000–8000 km of this type of driving.

Bergstra et al. [11] also performed low mileage accumulation tests via frequent cold short trips, which they identified as a particularly severe type of driving pattern, and they referred to these tests as the "Aunt Minnie" driving cycle. After 7000 miles of such driving, fuel dilution levels of 3–5% were found in the summer months and 8–11% in the winter months.

Schwartz [12] from General Motors installed a transparent window to investigate the state of the sump oil for cold, short trip driving conditions for gasoline fueled vehicles. It was found that such conditions led to an accumulation of fuel, water, and other contaminants that could cause increased wear, significant amounts of sludge to form, and fuel dilution levels of up to 10%. However, it was also found that simply driving longer distances and ensuring the engine was fully warmed up helped to drive off fuel and water levels, so that the lubricant performed well under these conditions.

It is worth commenting that in the "cold start" tests, quite a wide range of fuel dilution levels have been reported, ranging from as low as 5% to as high as 20%. The tests will have been performed on different vehicles, and at different temperatures (with some tests in summer, and some in winter, when different fuel formulations may have been used). The type of vehicle used will also affect the results, since the fuel injectors and engine clearances at low temperatures will differ from vehicle to vehicle. The main point to make is that driving from cold-start for short-trips will lead to substantial levels of fuel dilution in most vehicles.

Shayler et al. [13] developed an empirical model to predict fuel dilution levels. When applied to "Aunt Minnie" type driving patterns (very short trips from cold starts), the model predicted fuel dilution rates as high as 18–20%, in broad agreement with the findings of Kollman et al. [10] and Bergstra et al. [11]. For fully warmed up engine conditions, the model predicted fuel dilution levels to stabilize at around 2%.

Peralta (MIT) [14] found fuel dilution levels of 1–4% (by mass) in a fully warmed up Saturn four-cylinder gasoline engine. Similarly, Kovacs et al. [15] found typical fuel dilution rates of 1.2% in a fully warmed up single cylinder 398 $cm^3$ displacement Kohler engine. Frottier et al. (PSA/Peugeot/Citroen) [16] also found fuel dilution levels of 1.35% (by mass) in a Saturn engine after an 18-h long test.

Thomson et al. [17] developed a model to predict the process of absorption and desorption of fuel into and out of the lubricating oil films present in the piston ring zone and predicted equilibrium fuel dilution rates of 1.9% (by mass), broadly in agreement with levels measured in fully warmed up engines.

It should be noted that a number researchers also found that the fuel in the sump oil mainly consisted of the higher boiling point components of the fuel (the lighter components presumably having evaporated off). This effect was studied in detail by Murakami et al. [18] and also by Schramm et al. [19]. The important point to make is that the fuel components in the lubricant are not in the same proportions as those in the original fuel (and so the viscosity of the fuel components that accumulate in the lubricant is also likely to be different (higher) than the viscosity of the original fuel).

Most of these studies were carried out on older gasoline vehicles without aftertreatment devices fitted. More recent vehicles have a range of aftertreatment devices fitted to ensure tailpipe emissions compliance, and some of these devices inject extra fuel during their operation. An interesting recent paper by Tormos et al. [20] found temporary fuel dilution levels of over 20%, in a medium duty direct injection diesel engine, during the DPF (diesel particulate filter) regeneration mode (where the fuel post injection event is excessively delayed towards the expansion stroke).

## 4. The Viscosity of Fuel/Lubricant Mixtures

Zhmud [21] has recently reviewed how the viscosity of mixtures can be calculated. Essentially, if there are two fluids whose viscosities are $\eta_1$ and $\eta_2$ (these can either be dynamic viscosities in mPa.s or kinematic viscosities in cSt) which are present in a mixture in concentrations of $x_1$ and $x_2$ (and clearly $x_1 + x_2 = 1$), then the viscosity of the mixture, $\eta_{mix}$, is given by:

$$\log_e \eta_{mix} = x_1.\log_e \eta_1 + x_2.\log_e \eta_2 \tag{1}$$

Clearly, the mixture viscosity will be in the same units as the viscosity of the individual components (either in mPa.s or cSt).

Other authors such as Grunberg and Nissan have discussed adding additional terms to the right-hand side of Equation (1) as correction factors [22]

As an example of the use of Equation (1), consider gasoline at 40 °C. Table 2 shows that the kinematic viscosity of the fuel at this temperature would be approximately 0.7 cSt. Table 3 below shows the impact of fuel at different dilution rates on the viscosity of a typical SAE 5W-30 engine lubricant (whose viscosity at 40 °C will be approximately 55 cSt).

**Table 3.** Impact of fuel dilution rates on viscosity of lubricant/fuel mixture, from Equation (1), and assuming lubricant viscosity of 55 cSt and fuel viscosity of 0.7 cSt.

| % Fuel in Lubricant | Mixture Viscosity (cSt) |
|---|---|
| 0 | 55.0 |
| 1 | 52.65 |
| 2 | 50.40 |
| 5 | 44.22 |
| 10 | 35.55 |
| 20 | 22.98 |

## 5. Boiling Point Curves for Fuels

It is important to know the "distillation" curve (or "boiling point curve") of a fuel. This is essentially a plot of the amount of fuel that has evaporated versus temperature. These measurements are usually performed under standardized conditions, such as those described in ASTM D86 [23], or modifications thereof [24].

Typical distillation curves for RON 95 gasoline (RON = research octane number) and diesel are shown in Figure 1, which is a replot of data reported in reference [25].

The reason for the substantial difference in these curves is due to the composition of the fuel. Gasoline is a blend of hydrocarbons with carbon numbers between C4 and C11, whose boiling points lie between 25 and 210 °C. Many different types of hydrocarbons can be found in gasoline, including paraffins, iso-paraffins, olefins, aromatics, naphthenics, etc. Some oxygenated components and fuel additives are also present, and more recently various types of biofuels may be present (such as ethanol or methanol). On the other

hand, diesel fuels are obtained by a different refining process, and generally consist of hydrocarbons with carbon numbers in the range C10 to C16, whose boiling points lie in the range 160–360 °C. Fuel additives and various types of biodiesel components may also be present. Gasoline needs to be volatile enough to ensure easy start-up and good performance in cold climates, but not volatile enough to vaporize in the fuel tank or fuel lines. For diesel, if the volatility is too low, then this could lead to smoke formation, loss of power and higher fuel consumption. On the other hand, if diesel volatility is too high, then fuel vaporization could occur in the tank and fuel lines.

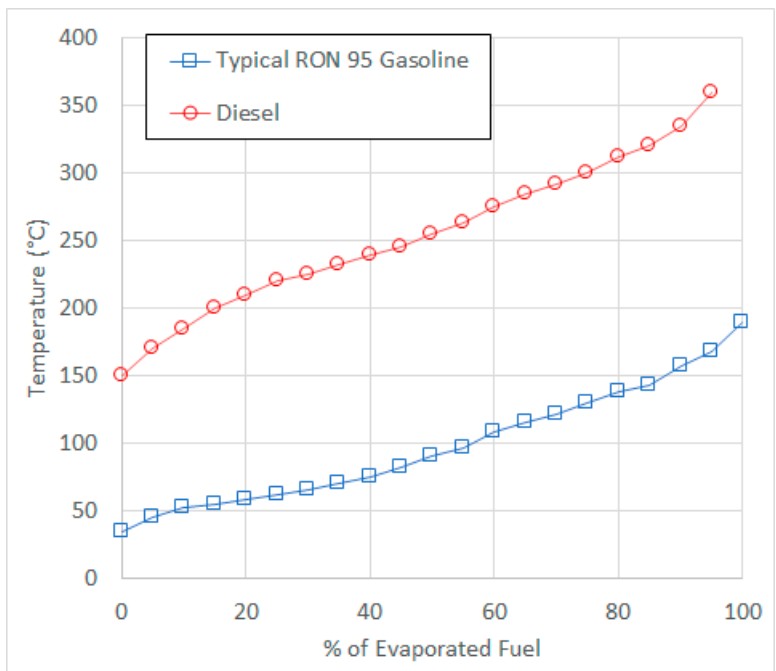

**Figure 1.** Typical distillation (or boiling point) curves for gasoline (blue) and diesel (red) (replot of data from reference [25]).

For gasoline vehicles, high levels of fuel dilution have been observed for short-trip, cold-start driving, but the gasoline that has built up in the sump will rapidly evaporate once the engine is fully warmed up and a longer duration journey is undertaken. On the other hand, if diesel fuel builds up in the lubricant sump, this will not evaporate quite so easily, and significant levels of diesel fuel dilution could still persist at sump oil temperatures in the range 100–150 °C.

## 6. The Impact of Fuel Dilution on Engines & Lubricants

Various researchers [26–47] have reported the type of issues that can occur due to excessive fuel dilution. These include an (1) increase in sump oil level, (2) change in lubricant viscosity, (3) sludge formation, (4) impact on friction and lubricant tribo-films, (5) low speed pre-ignition, and (6) removal of cylinder liner lubricant films. In practice, issues such as sludge formation and low speed pre-ignition are due to a combination of engine design, driving patterns, fuel quality, lubricant properties, as well as fuel–lubricant interactions. In practice, for a given engine, it is not usually possible to change the engine design, nor is it easily possible to change the typical fuel that is available in a specific geographical region. Therefore, it is often the lubricant manufacturer that is tasked with modifying the lubricant to address the problems that arise in these cases. These various effects are discussed more fully below.

### 6.1. Increase in Sump Oil Level

For a fuel dilution level of 5% and a sump that contains 4 L of lubricant, the total fluid volume in the sump would rise to 4.2 L. If the fuel dilution level is 20% (which is possible for drivers that only ever do cold-start, short trip, driving), then the total fluid level in the sump would be 4.8 L. These calculations do not include the effect of any evaporation of the lubricant. Many modern vehicles are fitted with an oil level sensor, and a warning light would appear on the dashboard if the oil level is too low or too high. For gasoline engine vehicles, a high oil level could in principle be reduced by simply driving on a longer journey (one or more hours) with a fully warmed up engine. Although this issue has been highlighted on the internet by consumers [26], OEMs [27], and oil companies [28], there does not appear to be any peer-reviewed scientific papers that have investigated this effect in detail.

### 6.2. Change in Lubricant Viscosity

For gasoline engines, fuel dilution will typically decrease the viscosity of the lubricant. This is more pronounced at low temperatures (up to around 50 °C) due to the fact the lubricant viscosity is substantially higher at low temperatures, and fuel evaporation losses will not be too significant at low temperatures. Table 3 showed that, for a typical SAE 5W-30 lubricant at 40 °C, a high fuel dilution level of 20% could reduce the lubricant viscosity from around 55 cSt to 23 cSt. However, at higher temperatures, gasoline fuel dilution levels will decrease, firstly due to evaporative losses, and secondly as lubricant viscosities are lower too. At 150 °C, even if fuel dilution levels were in the range 10–20% at 40 °C, they are likely to be reduced to less than 1% at 150 °C. At the same time, lubricant viscosities are likely to be around 3 mPa.s or so. If fuel viscosity is assumed to be 1 mPa.s, then the effect of fuel dilution, with the assumption that fuel dilution is only 1%, would be to reduce the viscosity to only 2.97 mPa.s.

For diesel engines, because of their higher boiling point range, less evaporation will occur compared to gasoline, and it is possible that higher levels of diesel fuel dilution could persist at oil temperatures in the range of 100–150 °C. This is now becoming more of an issue due to the increased use of fuel post-injection for the regeneration of aftertreatment systems (required to meet increasingly stringent tailpipe emissions regulations) [1,29].

Finally, it is worth noting that over long time periods, it is possible that fuel dilution could cause an increase in sump oil viscosity. This can happen if large molecular weight fuel additives (such as fuel detergents) accumulate, over time, in the sump oil. This is more likely to happen with highly additized fuels. Even though such additives are in the fuel at very low treat rates, a large amount of fuel is burnt during a typical oil drain interval. For example, if it is assumed that fuel detergents are only used at a treat rate of 0.1%, over the course of a typical oil drain interval (15,000 km), approximately 1000 L of gasoline will have been combusted. Even if only 10% of the fuel detergent additives end up in the lubricant, this would amount to around 0.1 L of a relatively high molecular weight component accumulating in the lubricant. For highly additized fuels that contain higher levels of fuel detergent additives, even higher amounts could accumulate. Over time, the accumulation of such additives could result in an increase in lubricant viscosity. It should be added that any increase in viscosity due to accumulation of higher molecular weight fuel additives is likely to be counteracted by the decrease in viscosity due to base fuel dilution of the lubricant. There has been very little published work on the impact of the accumulation of higher molecular weight fuel components in lubricants, and this is an area that would benefit from increased research effort.

It should also be added that monitoring the viscosity of a lubricant to detect fuel dilution is not straightforward for a number of reasons. Firstly, although lubricant viscosity can decrease as fuel dilution increases, the viscosity can also decrease due to permanent shear loss of viscosity modifier additives contained within the lubricant, and lubricant viscosity can also increase (due to accumulation of contaminants, such as soot, or due to oxidation of the lubricant). In addition, lubricant viscosity varies greatly with temperature,

and to a lesser extent with shear rate, so any measurement of lubricant viscosity needs careful control of these variables. Some researchers have compared viscosity measurements of fuel diluted lubricants, under carefully controlled conditions, and have found good agreement with laboratory measurements of fuel dilution [29].

### 6.3. Sludge Formation

Sludge formation in engines is often attributed to a combination of factors: (1) short-trip, stop-go driving style, (2) sump pan design, (3) type of lubricant used, (4) type of fuel used and (5) fuel–lubricant interactions. A number of studies on this topic were carried out in the 1990s, the most notable being papers by Murakami et al. [30] and Lillywhite et al. [31]. More recent studies on sludge formation have tended to focus on the impact of biofuels as reported in the review by Kurre et al. [32]. Murakami et al. [30] reported that NOx reacts with unburned gasoline (mainly olefins) to form sludge precursors. Driving conditions that favour high amounts of unburned gasoline (low temperature, stop-start driving with high accelerations) tend to cause more of the sludge precursors to accumulate in the lubricant. The rate at which these precursors cause sludge to form depends on the oxidative stability of the lubricant (a higher quality lubricant containing more antioxidants will take longer for sludge to form) and whether or not there are "low flow" areas in the sump pan. Lillywhite et al. [31] commented that the formation of sludgy deposits first became an issue in the early 1960s and again in the mid-1980s. Various engine design and lubricant formulation changes were undertaken to address the issues. Gasoline engine design was implicated as a major factor, particularly the blowby gas flow rates and the design of the crankcase ventilation system. Lillywhite et al. [31] also commented that lubricants with higher levels of antioxidants/detergents/dispersants could delay the onset of sludge well beyond the oil drain interval. Kurre et al. [32] reported that when water and metal are present in engine oil, lubricant antioxidants can be consumed rapidly, which can lead to corrosion, sludge, and varnish formation. In laboratory tests aimed at simulating lubricant sludge formation, poor quality fuels (often containing sludge precursors) are deliberately used to accelerate sludge formation.

Industry standard engine tests are in place to test the lubricant's ability to resist sludge formation. In the latest ILSAC GF-6 lubricant specification system (which is widely used in the USA and Asia), the Sequence VH engine test is used to evaluate the lubricant's ability to prevent engine deposit build-up (sludge and varnish). In Europe, for the latest ACEA light duty engine oil specifications, the CEC L-107–19 sludge deposit test is used and is commonly referred to as the M271 EVO test. (Note that ILSAC is the International Lubricant Standardization and Approval Committee, and ACEA is the European Automobile Manufacturer's Association –ACEA actually stands for the Association des Constructeurs Européens d'Automobiles). In the Sequence VH engine test, a 2013 Ford 4.6 L fuel injected eight-cylinder gasoline engine is used. The test duration is 216 h involving 54 cycles. To accelerate sludge formation, a fuel that contains sludge precursors is deliberately used, and engine blow-by levels are intentionally increased. At the end of the test sludge deposits are rated on the rocker arm covers, rocker arm cover baffles, timing chain cover, oil pan baffle, oil pan, and valve decks [33]. In the European M271 EVO sludge test [34], a Daimler M271 EVO engine is used, and a high temperature, high load phase is initially used to accelerate oil oxidation and to build up fuel in the sump lubricant. Then, a phase of lower speed, lower temperature testing is used, to encourage sludge formation. A special fuel containing sludge precursors is used. The amount of sludge in various parts of the engine including the oil pan is rated, and the amount must be below a certain limit for the lubricant to pass the test.

### 6.4. Impact on Friction and Lubricant Tribo-Films

Unburned fuel can affect friction in two ways. Firstly, as described earlier, fuel components that accumulate in the lubricant can affect lubricant viscosity. Since an engine is predominantly lubricated hydrodynamically, a change of viscosity will directly change

friction in components such as the journal bearings and the piston assembly. For gasoline engines, a lowering of viscosity would generally be expected, and so this would usually lead to lower engine friction. For gasoline engines, the highest impact of fuel dilution is at lower temperatures, and so fuel dilution can substantially reduce friction during the engine warm up phase. At higher temperatures, the gasoline contained in the lubricant will usually evaporate off, and so will not tend to cause wear issues. On the other hand, for diesel engines, significant amounts of diesel fuel could remain in the sump oil at elevated temperatures (more than 100 °C) and so there is concern that diesel fuel dilution could lead to lower viscosities that could lead to thinner oil films and wear issues (in components such as journal bearings) at higher temperatures. Research into this is currently ongoing [1]. The second way in which friction can be affected is if the fuel, or the fuel additives, interfere with the action of lubricant additives (such as friction modifiers or anti-wear additives). For example, Costa and Spikes [8] have reported that ethanol (widely used as a biofuel in gasoline) significantly reduced the thickness of the ZDDP anti-wear tribo-film that forms in highly loaded lubricated contacts. This will tend to reduce the friction in the contact but will also likely lead to higher levels of wear. Other relevant works in this area include those of Lee et al. [35,36], Björling et al. [37], and Notay et al. [38].

*6.5. Low Speed Pre-Ignition*

In recent years, the phenomenon of low-speed pre-ignition (LSPI) has become more commonplace. LSPI is a premature combustion event, that occurs randomly and infrequently, prior to spark ignition in turbocharged, downsized gasoline engines, which can result in extremely high cylinder pressures, leading to loud knocking noises and potentially catastrophic damage to the piston rings and piston. This is a phenomenon that depends on engine design, fuel quality, and lubricant composition. In practice, it is difficult to change engine design, and fuel quality primarily depends on the country you are in, so most efforts have focused on modifying the lubricant formulation to reduce LSPI. Research is ongoing into both lubricant and fuel properties on LSPI occurrence [39] and the impact of fresh and aged lubricants [40]. Work has also been published to assess the impact of engine design [41]. Onodera et al. [42] have reported that reformulation of the lubricant (by replacing calcium detergents with magnesium detergents and using an increased dose of molybdenum-based friction modifiers) can result in lower LSPI occurrence. The impact of base oil viscosity and base oil quality has also been investigated [43]. Andrews et al. [43] reported that engine oils formulated with higher viscosity base stocks produced more LSPI events. The impact of fuel properties on LSPI has also been investigated [44,45]. Jatana et al. [44] tested four different fuels in an engine run at identical LSPI prone operating conditions. They found that fuels with similar boiling properties and octane numbers exhibited similar LSPI number counts, but there were vastly different LSPI magnitudes and intensities. Their results highlighted that fundamental fuel properties such as flame speed are critical to characterizing both LSPI propensity and intensity. More recently, Swarts et al. [45] investigated the impact of market fuels on LSPI, and tested fuels with a range of properties (composition, boiling point distribution, ethanol content, and particulate matter index (PMI)). Their tests used a 2-L GM LHU engine running high-load, low-speed, steady state tests. It was found that the PMI and certain boiling points of the fuel correlated best with the frequency of LSPI events. The authors also found that decreased LSPI severity corresponded with increased octane numbers and higher ethanol content of the market fuels. A number of published studies have investigated fuel lubricant interactions on LSPI [46–48]. Hu et al. [46] found that the properties of oil particles entering the engine cylinder were significantly affected by fuel dilution. Their work used a highly boosted 1.8 L turbocharged gasoline direct injection (TGDI) engine fueled with RON93 gasoline. Dilution of the engine oil by the fuel lowered the boiling point and auto ignition point of the oil (compared with oil that did not have fuel dilution). The authors claimed that the fuel diluted oil particles could easily form self-ignitable gaseous mixtures that could help to trigger low speed pre-ignition. The authors claimed that the frequency of

LSPI was strongly linked to the minimum auto ignition temperature of the oil particles. Kocsis et al. [47] investigated LSPI when the fuel and oil properties were varied at the same time. The aim of the study was to investigate whether a low LSPI activity lubricant could suppress the increased LSPI from a high LSPI fuel, and vice versa. The authors found that a low LSPI activity fuel was relatively insensitive to the lubricant used in the tests (in which a 2.0 L GM Ecotec LHU engine was used) whilst a high LSPI activity fuel could be moderated by using a low LSPI activity lubricant. As expected, the combination of a high LSPI activity fuel and a high LSPI activity lubricant resulted in large numbers of LSPI events. Kar et al. [48] found that the oil composition had a strong impact on LSPI frequency and that LSPI frequency could be reduced by changing the lubricant formulation. In addition, it was found that fuels blended with high polyaromatic content increased LSPI frequency significantly and also caused a significant increase in particulate mass and particulate number emissions.

Clearly, research into LSPI is still very much active, but there is a clear picture emerging that fuel properties, lubricant properties, and engine design all play a part, and numerous papers have suggested that the interaction between the fuel and lubricant is also important in the understanding of LSPI.

### 6.6. Washing off of Lubricant Film on Piston Liner Wall

Excessive fuel dilution can wash lubricant films off piston cylinder walls, which will potentially negatively impact piston ring lubrication. This has been more of an issue in gasoline direct injection engines, which can partly be alleviated by redesigning the angle at which fuel is injected into the combustion chamber. Hu et al. [49] report that wall wetting can be caused by direct impingement of fuel sprays onto the cylinder wall or can occur indirectly. It was reported that the unburned fuel can (1) dilute the concentrations of lubricant additives (anti-wear, corrosion inhibitors, antioxidants, dispersants, detergents, etc.), (2) can potentially react with some oil additives and reduce their functionality, and (3) reduce the oil's viscosity, making the oil more volatile, potentially increasing cylinder wear and oil consumption (since the more volatile oil is carried away in blowby gases in greater concentrations). Increased wear in GDI engines due to wall wetting by the fuel is also mentioned by Chincholkar et al. [50] and Quieroz et al. [51].

## 7. The Impact of Biofuels on Lubricant Performance

There has been much research into the impact of biofuels on engine performance and the impact of biofuels on lubricants and lubrication [52–68].

For gasoline engines, many countries use ethanol as a biofuel, often at concentrations up to 5% (E5) or up to 10% (E10). Most modern vehicles/engines manufactured since 2011 can use such fuels without any engine modifications required. In other countries such as the USA, much higher concentrations of ethanol are used in some states (such as E85, where 85% of the fuel is ethanol) and in Brazil E100 has been used as a fuel since the 1970s. For such high ethanol concentrations, engines need modification to enable them to run on such fuels. In China, methanol is used in some regions and can be blended into fuel at levels ranging from 5% to 85%. China has a national quality standard for methanol blends of 85% and a national standard for a 15% blend of methanol (M15) in gasoline is pending approval from the Chinese authorities.

It should be noted that the available evidence and experience to date suggests there are no significant lubrication issues in service from the use of ethanol up to a concentration of 10% in gasoline, for modern vehicles.

In a recent review, Khuong et al. [59] reported that bioethanol dilution has a significant effect on the properties of automotive lubricants, particularly on oil consumption, corrosion, wear, and sludge. The authors also noted that ethanol can attract water, potentially resulting in emulsions of ethanol/water/lubricant/lubricant additives that could be a precursor for sludge formation. It was also noted that, as bioethanol dilution rates increased, the total base number of the lubricant decreased, and the total acid number increased (i.e., the

lubricant became more acidic). Boons et al. [54] carried out vehicle field trials to investigate the impact of E85 on lubricant performance. They reported that ethanol can be aggressive on metals and seals, and could potentially cause increased corrosion, rusting, wear, and sludge. It was found from the field trials that the use of E85 could lead to significantly higher water levels in the lubricant (compared to E10 fuel). With these high water levels, oil water micro-emulsions were formed at low ambient temperatures. However, despite these high water levels (and high levels of ethanol in the lubricant) no engine or driving issues were observed in the field test, and no separation was observed with any of the test oils. At higher outside temperatures the water and ethanol levels in the lubricant very quickly dropped to low levels. The authors also commented that the use of E85 did not lead to higher valve train rust levels. De Silva [60,61] reported results from both tribology bench tests, and single cylinder engine tests run on a range of fuels (unleaded RON 95 gasoline, E10, E20 and E85) with a high-quality SAE 5W-30 lubricant (that met API SL/CF and ACEA A3/B3/B4 specifications). De Silva [61] reported that lubricant films present on the piston skirt and along the cylinder liner were susceptible to ethanol dilution under cold-start and warm-up driving conditions, particularly at low loads. However, once the engine was fully warmed up, the temperature in the piston assembly was higher than the boiling point of ethanol. Fluid samples were taken from the cylinder liner to better understand it's composition. Analysis of the FTIR spectra of the samples found that there were varying concentrations of lubricant, ethanol, and water depending on the fluid extraction point on the liner, the type of fuel used, and the air fuel ratio (AFR) at which the engine was operated. For these particular tests (which started from cold) it was found that ethanol and water contamination was much higher at top dead centre compared to mid-stroke and bottom dead centre positions. Even though ethanol was only used in the fuel at the 10 or 20% level, between 1–2% of ethanol was found in the TDC lubricant film samples. De Silva [60] also reported friction measurements from a Plint TE77 reciprocating tribometer using gasoline engine lubricant contaminated with ethanol and water. The contaminated sample separated into two distinct phases, an oil phase and a water and ethanol based "white sludge" phase, and the friction of both phases was measured. It was stated that some lubricant additives were preferentially absorbed into the "white sludge" phase, which could cause a reduction in viscosity of the oil phase (due to a loss of viscosity modifier additives). Significant reductions in friction were found for both phases (compared to the original uncontaminated lubricant).

Hurst [67] investigated the detailed chemical mechanisms leading to lubricant degradation from ethanol fuel dilution. In comparison to biodiesel, Hurst reported that the oxidative stability of model lubricants (containing detergents, dispersants and a range of antioxidants) was enhanced by ethanol fuel dilution and subsequent evaporation. This was explained by lubricant additives preferentially being absorbed by the ethanol, and then forming reverse micelles (this phenomenon was confirmed by light scattering measurements–the undiluted model lubricant sample was clear whereas the ethanol contaminated sample was "hazy", which was attributed to the inhomogeneous reverse micelles).

For diesel engines, biodiesel is used as a biofuel in concentrations up to 20% (known as B20). In some countries lower amounts of biodiesel are blended into diesel fuel (a 5% blend is denoted by B5). Biodiesel can be made from nearly any feedstock that contains adequate free fatty acids. Example feedstocks are: vegetable oils, used cooking oils, yellow grease and animal fat. Different types of vegetable oils are used in different geographical regions of the world. In the USA, soybean oil is mainly used, along with corn oil and canola oil. In other countries alternative vegetable oils used include rapeseed oil, sunflower oil, jatropha, and palm oil. Biodiesel is produced through transesterification, a chemical process that converts fats and oils into fatty acid methyl esters (FAME). There have been vehicle issues with biodiesel from some feedstocks, whereby fuel filters have been blocked, and the low temperature properties of the fuel have been adversely affected. In addition, in some cases, biodiesel fuel dilution has adversely affected the lubricant oxidation performance, leading to recommendations by OEMs for shorter oil drains when those biodiesel fuels are used.

Moreover, 200-h tests using a single cylinder diesel engine run on B20 diesel fuel blends [65] found that the use of B20 fuel resulted in a reduction in lubricant viscosity and an increase in the acidity of the engine oil. Chemical analysis of the lubricant also found an increase in fuel residue, increased corrosion, and increased oxidation of the engine oil. An 18-month field test on a fleet of buses that used B100 biodiesel [64] found that replacing traditional diesel fuel with B100 diminished the engine lubricant drain period from 20,000 km to 13,000 km (for mono-articulated buses) and from 15,000 km to 10,000 km (for bi-articulated buses). Richard et al. [63] reported that the use of biodiesel (at B20 levels) resulted in worse corrosion performance of diesel engine oils. It was found that the number of double bonds in the fatty acid chain correlated with the FAME induced corrosion (i.e., the higher the number of double bonds, the higher the degree of corrosion). However, it was also reported that increased use of corrosion inhibitors in the oil (to protect against both copper and lead corrosion) could be used to bring corrosion back to acceptable levels. Researchers from Infineum [68] have also reported on the impact of B20 fuel in a 100,000-mile field trial using medium duty buses in Las Vegas. In these trials, high biofuel dilution levels of 10–50% were observed and this was attributed to (1) the design of the in-cylinder post fuel injection (for regeneration of the diesel particulate filter), (2) engines that were not specifically designed to run on biodiesel, and (3) the extreme stop and go nature of the driving cycle, which also included extensive idling and a lack of highway speeds. Despite the high biofuel dilution levels, at the end of the trial, all engines showed excellent sludge control and no issues with cylinder liner wear. Higher bearing wear was seen for engines running on B20, although the authors claimed that the use of high quality oils could offset this fuel effect. The authors also commented that oil drains may need decreasing (to counter the increasing acidity due to the use of B20) and that the viscosity decrease due to fuel dilution needed to be monitored.

## 8. The Impact of Fuel Additives

Nowadays, fuels sold in many countries include various additives (at relatively low treat rates), such as deposit control additives (fuel detergents), corrosion inhibitors, cold flow improvers, lubricity additives, and in some cases friction modifiers. A comprehensive review of fuel additives was published by Bennett [69] in 2014.

Fuel detergents can keep fuel injectors free of deposits and prevent deposits forming in the combustion chamber. In some cases, if injectors are already dirty, and there are existing combustion chamber deposits, it has been claimed that the use of the detergent additized fuel can "clean up" the fuel injectors and the combustion chamber, bringing the engine performance back to that when the engine was new. These additives became more important once direct fuel injection engines became more commonplace (in older port fuel injected engines, fuel would flow over the engine valves and "wash off" any deposits that were accumulating).

Friction modifiers first started to be added to fuels in the late 1990s. Hayden [70] claimed that 25% of incoming fuel additives impinge on the cylinder wall and can be captured by the thin oil film. It was argued that since the cylinder wall surface temperatures are less than 175 °C, the additives could survive the combustion process. Hayden [70] claimed that, via this mechanism, a gasoline friction modifier fuel additive would be delivered to the cylinder wall and could help to reduce friction of the uppermost piston rings. The fuel economy benefits of the friction modifiers were measured in the Sequence VI engine dynamometer fuel economy test, and it was claimed that there was both an instantaneous benefit (presumably from friction reduction in the piston assembly) and also a longer-term effect (assumed to be due to the accumulation of the gasoline friction modifier in the lubricant and subsequent friction reductions in other engine components, such as the valve train). The authors claimed that the use of friction modifiers in the lubricant did not negate the fuel economy benefits from fuel borne friction modifiers. The treat rate of the gasoline friction modifier was quoted as being 260 pounds per thousand barrels (which works out to be around 0.087%, assuming a barrel is 300 pounds), although lower

treat rates of 80 pounds per thousand barrels (0.027%) and 20 pounds per thousand barrels (0.007%) were also tested. The fuel economy benefits observed increased with higher additive treat rates. The authors commented that, although a fuel economy benefit was seen from the gasoline friction modifiers in the Sequence VI engine test, Sequence VI-A and VI-B fuel economy engine tests were much less responsive to the presence of the gasoline friction modifier (this suggests any benefit from gasoline based friction modifiers will vary depending on engine design, with engines that have more mixed/boundary lubrication being more likely to have a benefit from such additives–the sequence VI engine had sliding valve trains, whereas the engines used in the VI-A and VI-B engine tests had roller follower valve trains, which may explain the differences seen in the work). Since the amount of mixed/boundary lubrication can vary greatly with engine design (as recently reported by Taylor et al. [71]) any benefit from the use of gasoline friction modifiers is likely to be very vehicle dependent. Hayden et al. [70] did not disclose the chemistry of the friction modifiers used, although a later patent application by the same authors suggests the gasoline friction modifiers used were ester based [72]. Shell researchers [73] investigated the impact of the carrier fluid on vehicle acceleration. Usually, active additives are supplied in a base oil carrier fluid. A Ford Zetec engine, installed on a dynamometer, was tested with fuels containing different treat rates and viscosity grades of base oil. Somewhat surprisingly, it was found that the measured acceleration benefit increased as the viscosity of the base oil increased, with a maximum benefit of just over 10% improvement in acceleration time when HVI-650 mineral base oil (commonly known as brightstock, with a kinematic viscosity of 32 cSt at 100 °C) was used at a treat rate of 2%. Figure 2 plots the data contained in the patent [73]. It was suggested that the effect could be due to (1) unburned base oil acting as an extra lubricant at the top piston ring, reducing the amount by which the top piston ring is starved, and substantially reducing the friction there, or (2) the base oil in the fuel may be affecting the indicated mean effective pressure (IMEP) of the fuel. The conclusion from this work is that some of the benefits of gasoline friction modifiers may be from the carrier fluid, rather than the active additive. Smith [74] also investigated the impact of friction modifiers in the fuel lubricant mixture at the top of a piston ring. Smith [74] found that the addition of environmentally friendly friction modifiers administered to the engine via the gasoline could lead to fuel economy improvements of approximately 2% (as measured in a single cylinder Ricardo engine test). Recently, researchers have also investigated the use of boric acid as a fuel additive for reducing friction [75,76].

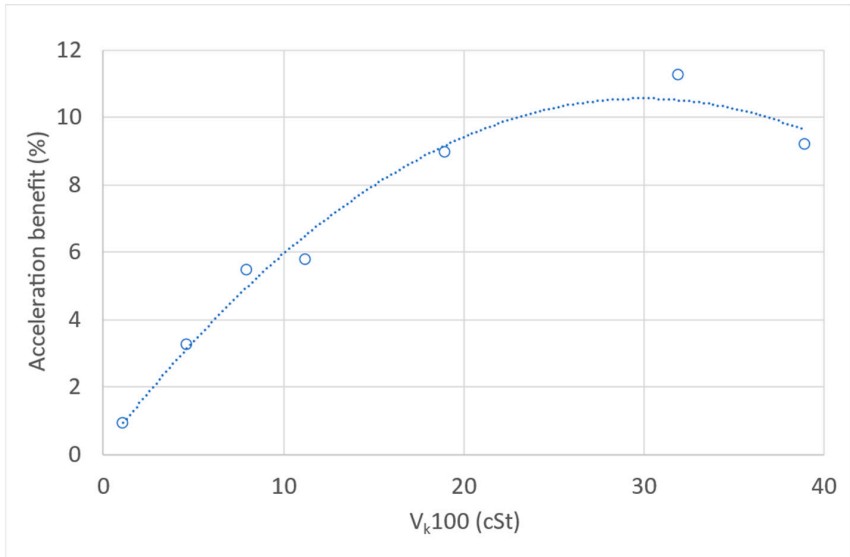

**Figure 2.** Acceleration benefit (%), as measured in a Ford Zetec engine, for fuels containing 2% of base oils of different viscosity grades (the kinematic viscosity of the base oils at 100 °C is on the horizontal axis) compared to the baseline fuel [73].

### 9. Recent Technology Impact on Fuel-Lubricant Interactions

Two recent technology trends will impact fuel dilution. Firstly, more stringent tailpipe emissions limits (on NOx and particulate matter) are leading to the widespread use of particulate filters (mainly for diesel cars and heavy-duty trucks). If these emissions limits are tightened in future years, gasoline particulate filters may need to be introduced for some gasoline engines too (in fact, in some geographical regions gasoline particulate filters are already being used). Active regeneration of the particulate filters, by post fuel injection (which burns off soot and avoids plugging of the particulate filters) is leading to reports of high fuel dilution levels (of the order of 10–20%), particularly for stop-start, delivery type driving conditions (and these driving conditions could also occur in congested city centres). These issues are becoming of more importance for the latest emissions regulations (e.g., Euro VI, Bharat Stage VI emissions standards in India, and China VI emissions standards). Hermann [77] has reported that a single post-injection particulate filter regeneration event can lead to a temporary fuel dilution level of 8.5%, whereas a multiple post-injection event can lead to a temporary fuel dilution level of 11%. These numbers are consistent with those reported by Tormos et al. [20].

The second technology that will impact fuel dilution of lubricants is that of vehicle electrification [78]. There are increasing numbers of hybrid electric vehicles being manufactured that have both a conventional engine and a battery. The conventional engine in such vehicles will potentially run infrequently and be subjected to many more stop-starts than found in conventional vehicles. This is likely to lead to lower overall oil sump temperatures, and increased fuel dilution levels. Fan et al. [79] have recently reported the results of a field trial which used hybrid and conventional Geely vehicles equipped with a 1.5 L turbo-charged direct injection (TGDI) engine, running on different driving cycles. For the hybrid vehicles operating on multiple repeats of the European ECE-15 driving cycle, the hybrid vehicle had a lower operating oil temperature compared to the conventional vehicle (80 °C versus 95 °C) and the fuel dilution level was higher (at just over 3% for the hybrid vehicle after twenty ECE-15 driving cycles, compared to only 1.5% fuel dilution for the conventional vehicle).

### 10. Conclusions

This review has attempted to summarize the current state of knowledge of fuel–lubricant interactions. Fuel–lubricant interactions are becoming more important due to the increased number of hybrid cars on the road and increasingly stringent emissions legislation, leading to the use of active regeneration particulate filters. Fuel dilution of the lubricant can affect lubricant viscosity, which can impact friction and wear, and can lead to increases in the sump level (which can result in warning lights appearing on car dashboards). Fuel dilution of the lubricant can also result in sludge formation and low speed pre-ignition problems. Gasoline fuel dilution is perhaps less of a concern, since the high volatility of gasoline means that, at higher operating temperatures (100–150 °C), most of the gasoline in the lubricant will evaporate off, although it is possible some of the higher molecular weight fuel additive components (such as fuel detergents) could continue to accumulate in the lubricant over the oil drain interval. What is more concerning in recent years has been the higher levels of fuel dilution seen in diesel vehicles. The lower volatility of diesel fuel means there could still be relatively high levels of diesel fuel present in the lubricant at relatively high temperatures (100–150 °C). In addition, much work has been undertaken to better understand the impact of biofuels on engine performance. The use of ethanol and methanol in gasoline cars can potentially lead to increased water ingress into the lubricant, sludge issues, and potentially more corrosion. The use of biodiesel will also potentially lead to decreased oil drain intervals. It is anticipated that current concerns about fuel dilution will lead to increased research activity in this area in the near future.

**Funding:** This research received no external funding.

**Conflicts of Interest:** The author declares no conflict of interest.

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
