# Peer review of "Fuel-Lubricant Interactions: Critical Review of Recent Work"

_lubricants, doi:10.3390/lubricants9090092_

Round 1

Reviewer 1 Report

This manuscript is an excellent review of the current knowledge of fuel-lubricant interactions. In addition to summarizing the literature on these phenomena in gasoline and diesel engines, the author describes commonly used concepts and mentions standardized testing methods, which is very helpful for the reader. This manuscript can be accepted for publication in its current form. One small typo on p. 190 ("whole" should be "whose") can be corrected in proofs.

Author Response

Many thanks to the reviewer for their comments and interest in the paper. The typo highlighted in the review has been corrected. 

Reviewer 2 Report

The reviewer finds the article interesting and well written. However, before publication, the reviewer wants to highlight a few points for the author to consider:

Section 2, page 2, rows 53-55. ” Additional measurements of the viscosity of diesel fuels were reported by Schaschke et al in 2013 [4]. This data also included measurements at high pressures. For measurements at atmospheric pressure, their measured dynamic viscosities ranged from about 3.2 mPa.s at 25C to 0.97 mPa.s at 100C.” Here it would be appropriate with a comment from the author on how these measurements compare to those earlier discussed in connection to reference 2 and 3.

Section 3, page 3, rows 102-115: “Kollman et al [10] carried out a series of “SNAIL” field trials, in which a number of different gasoline cars were only ever driven short distances (up to 10 km) from a cold start. For this type of operation, fuel dilution in the sump oil was typically in the range 10- 20% after 4000-8000 km of this type of driving.
Bergstra et al [11] also performed low mileage accumulation tests via frequent cold short trips, which they identified as a particularly severe type of driving pattern, and they referred to these tests as the “Aunt Minnie” driving cycle. After 7,000 miles of such driving, fuel dilution levels of 3-5% were found in the summer months, and 8-11% were found in the winter months.
Schwartz [12] from General Motors, installed a transparent window to investigate the state of the sump oil for cold, short trip driving conditions for gasoline fueled vehicles. It was found that such conditions led to an accumulation of fuel, water, and other contaminants, that could cause increased wear, significant amounts of sludge to form, and fuel dilution levels of up to 10%.”

There is a massive difference between these references. For the readers convenience, it would be nice of the authors could mention if Kollman et al [10] and Schwartz [12] were performed in summer or winter months. Although this information is maybe also more or less useless since this could vary much in different countries as well.

Section 5, page 5, row 188-190: “On the other hand, diesel fuels are obtained by a different refining process, and generally consist of hydrocarbons with carbon numbers in the range C10 to C16, whole boiling points lie in the range..” I suppose that whole should be whose?

Section 6.2. Just as a point of discussion. If 15000 km of driving is assumed to lead to 10 % of fuel detergent additives to end up in the lubricant thus causing an increase in viscosity. Is it not reasonable that this would be counteracted by the fuel dilution that will be necessary for 10 % of the fuel detergent additives to end up in the lubricant? Also, is 10 % a value on the low side? Is there any references where this has been explored?

Section 6.4, page 8. As this section is somewhat low on references, the author may consider adding some of these. They are not solely focused on fuel dilution or fuel additives, but they discuss these as well:

Lee PM, Priest M, Stark MS, et al. Extraction and tribological investigation of top piston ring zone oil from a gasoline engine. Proc Inst Mech Eng J: J Eng Tribol2006; 220(3): 171–180.

Lee PM, Stark MS, Wilkinson JJ, et al. The degradation of lubricants in gasoline engines: Development of a test procedure to evaluate engine oil degradation and its consequences for rheology. Tribol Interf Eng Ser 2005; 48:593–602

Björling, M.; Berglund, K.; Spencer, A. & Larsson, R.
The effect of ageing on elastohydrodynamic friction in heavy-duty diesel engine oils
Proceedings of the Institution of Mechanical Engineers, Part J: Journal of Engineering Tribology, 2016

Rai Singh Notay, Martin Priest, Malcolm F. Fox,
The influence of lubricant degradation on measured piston ring film thickness in a fired gasoline reciprocating engine, Tribology International, Volume 129, 2019 ,Pages 112-123,

Section 8, page 12, rows 571-572: “Smith [70] also investigated the impact of friction modifiers in the fuel lubricant mixture at the top of a piston ring.” Maybe a short comment on the main findings of this article could be added here.

The author may consider adding the findings of this work to section 8:

Larsson, E., Olander, P., Jacobson, S.
(2018). Boric acid as fuel additive - Friction experiments and reflections around its effect on fuel saving.
Tribology International, 128: 302-312

Larsson, E., Olander, P., Jacobson, S.
(2017).
Boric acid as a lubricating fuel additive - Simplified lab experiments to understand fuel consumption reduction in field test.
Wear, 376: 822-830

Section 9, page 13. Rows 582-586: “Two recent technology trends will impact fuel dilution. Firstly, more stringent tailpipe emissions limits (on NOx and particulate matter) are leading to the widespread use of particulate filters (mainly for diesel cars and heavy-duty trucks). If these emissions limits are tightened in future years, gasoline particulate filters may need to be introduced for some gasoline engines too.” As far as the reviewer is aware also gasoline engines require particulate filters to pass even todays emission limits.

Author Response

Many thans for the reviewer's comments and interest in the paper. 

  1. I have added some comments in Section 2 about how the measurements of Schaschke et al relate to the measurements of others on diesel fuel viscosity.
  2. I agree with the reviewer that there is quite a wide range of results that have been reported for fuel dilution under cold start conditions. Fuel dilution can be as low as 5% and as high as 20%. Results depend on whether tests were carried out in summer or winter (as some regions use different fuels in different seasons) and there will also clearly be a vehicle effect. I have added some extra comments in this section to aid the reader. 
  3. The typo found in section 5 has been corrected.
  4. Section 6. The reviewer is correct that any viscosity increase from the accumulation of fuel detergent additives in the lubricant could well be masked by a decrease in lubricant viscosity from fuel dilution. The calculation assumed that only 10% of the fuel detergent ended up in the lubricant sump, which could well be an underestimate. The difficulty here is that fuel formulations are not publicly available, so the amount of fuel detergent additive used in the fuel is not known (although is likely to be less than 0.1%) and it is not known how much of the additive survives the combustion process and ends up in the sump. Fuels are available that have higher additive treat rates, which presumably have higher levels of such additives. I am not aware of published work in this area, and so have added a comment in this section to suggest this could be a useful area for future research. 
  5. The references supplied by the reviewer for section 6.4 were very useful and have been added
  6. In section 8 I have added some further comments on the findings of Smith et al, and have mentioned the additional references provided by the reviewer. 
  7. In section 9, I note the reviewer's comments. My impression is that in some geographical regions, some gasoline vehicles do need to be equipped with gasoline particulate filters, but that it is ususally the higher powered gasoline vehicles that need such filters at present. I have reworded the paragraph appropriately.

Reviewer 3 Report

Interesting and informative study.  No new work presented, as the title indicates, but interesting nevertheless.

Author Response

Many thanks to the reviewer for their comments and interest

Reviewer 4 Report

This is a review paper on the subject of the recent work on fuel lubricant interactions which means that gathering comprehensive information and literature of the fuel lubricant interactions and then comparing the obtained results and interpreting them together with a comprehensive discussion. After a careful peer-reviewing process of this critical review I have not found this manuscript in this way and what I found is non-consistence information from different papers that just re-written as a collection of information. A critical review paper must debate the different aspects of the issue, combine and compare the results in different aspects and points of view and plot them. While here, only two plots from others were shown again. The advantages and disadvantages of “biofuels” and their comparison should be listed in this critical paper. Also, a further complication of the lack of scientific understanding is the problem in the organization and logic of this critical review manuscript. Instead of dealing with important points individually and specifically, the authors mixed many things throughout the paper and, therefore, failed to provide a clear and convincing discussion. Therefore, this manuscript cannot be accepted in the “Lubricants” journal in the present form. There are also numerous mistakes or meaningless, vague statements in the paper. Some examples are given below:

  1. The title of the review paper must be improved to highlight the overall subject of this study. It is not acceptable in the present form. Please pay attention that, this is not a book chapter or not an individual technical report.
  2. It is beneficial to provide a separate timeline-based figure of the recent investigations on the subject of fuel-lubricant interactions. Providing step by step or a follow-up flowchart or something like this can improve the quality and readability of this review in the future.
  3. The main issue with the paper in its present form is the introductory part, which needs to be expanded. The “Introduction” section is not acceptable and should be more expanded as a separate chapter. This is particularly important for a comprehensive critical review, which should have an introduction aimed at a broad audience, providing an accessible introduction to the topic for a general reader (especially focused on the recent advances in fuel-lubricant interactions).
  4. The “Viscosity of Fuels” and “The Impact of Fuel Additives” sections must be substantially improved for increasing the quality of the critical review paper. Authors are encouraged to expand the main characteristics which have been collected as the results in these sections.
  5. Furthermore, the review should provide an intellectual contribution/insight/outlook beyond what is already covered in existing reviews on the same or closely related topics. It should be explicitly clear in the paper (usually but not always in the introduction) how this is achieved. For example, the critical review should provide a distinct contribution from reviews on “Recent Technology Impact on Fuel-Lubricant Interactions”, as well as the effect of other parameters such as fuel additives, viscosity, and so on. Making the distinction clear is easy, but just remember that you are writing also with non-specialist readers and students new to the topic in mind – be explicit and clear on the difference.
  6. This review manuscript should provide a critical assessment of the topic and the literature and be carefully fact-checked. A literature survey on the subject of biofuels and their properties alone is not sufficient for this case.
  7. Also, it is important that the article is written in a pedagogical and logical manner with a clear connection between sections and different parts of the paper. As can be seen, the submitted manuscript lacks a clear division between the sections. For example: “2. Viscosity of Fuels” and “3. Fuel Dilution Levels in Vehicles” sections which have not a direct relationship together. One is related to the fuel and another is related to the vehicle. A substantial re-organization based on the important factors is beneficial for this review.
  8. The references format must be double-checked by the authors. New references are more beneficial for this case due to the improvement of the quality and readability of this paper.
  9. Providing more information about the studies on the fuel interactions is required as figures, graphs, illustrations, schematics, and tables. Please revise. Also, remission for each table/figure must be provided by the authors.

Author Response

Many thanks to the reviewer for their comments. 

I should point out, firstly, that I am not aware of any other reviews on fuel-lubricant interactions that are available in the scientific literature (if the reviewer is aware of any please let me know). 

Secondly, I would argue that the paper is written in a logical manner. It is known that fuel mixes with the lubricant in engines, and as viscosity is a key property of lubricants, how viscosity is changed by fuel accumulation is of great interest. Therefore I start with a section on viscosity measurements of fuels (this cannot be expanded much further as there are not that many scientific papers reporting such measurements). I then look at how much fuel gets into the lubricant. Then I look at blending rules for predicting the viscosity of the mixture. I then expand on the other aspects of fuel-lubricant interaction (sludge formation, impact on friction, LSPI, etc.) and the impact of biofuels. The reviewer may not like how I have written the paper, but I believe many researchers in the field will regard the way it is written as being entirely logical. 

I do not understand why the introduction is not acceptable to the reviewer. It is a precis of the content of the rest of the paper and none of the other reviewers had a problem with it. 

The section on fuel additives cannot be expanded too much because very little research has been published on the impact of fuel additives that accumulate in the lubricant. In addition, of course, fuel formulations are not in the public domain, and so additive treat rates and chemistry are not easy to obtain (or publish). 

Round 2

Reviewer 4 Report

Accept